# Multifaceted Analysis of the Environmental Factors in Severely Injured Trauma: A 30-Day Survival Analysis

**DOI:** 10.3390/healthcare11091333

**Published:** 2023-05-05

**Authors:** Sung Woo Jang, Hae Rim Kim, Pil Young Jung, Jae Sik Chung

**Affiliations:** 1Trauma Center, National Medical Center, Seoul 04564, Republic of Korea; longvoiceactor@nmc.or.kr; 2College of Natural Science, School of Statistics, University of Seoul, Seoul 02504, Republic of Korea; haley2203@uos.ac.kr; 3Department of Traumatology, Department of Surgery, Wonju Severance Christian Hospital, Yonsei University Wonju College of Medicine, Wonju 26426, Republic of Korea; surgery4trauma@yonsei.ac.kr

**Keywords:** trauma centers, risk factors, survival analysis

## Abstract

(1) Background: Most factors that predict the in-hospital survival rate in patients with severe trauma are patient-related factors; environmental factors are not currently considered important. Predicting the severity of trauma using environmental factors could be a reliable and easy-to-use method. Therefore, the purpose of this study was to determine whether environmental factors affect the survival in patients with severe trauma. (2) Methods: Medical records of patients who activated trauma team in the single regional trauma center, from 2016 to 2020, were retrospectively analyzed. After exclusion of young patients (<19 years old), cases of mild trauma (ISS < 16), and non-preventable deaths (trauma and injury severity score <25%), a total of 1706 patients were included in the study. (3) Results: In the Cox proportional hazard regression analysis, older age, night compared with day, and high rainfall were identified as statistically significant environmental predictors of mortality due to severe trauma. The relationship between mortality and precipitation showed a linear relationship, while that between mortality and temperature showed an inverted U-shaped relationship. (4) Conclusions: Various environmental factors of trauma affect mortality in patients with severe trauma. In predicting the survival of patients with severe trauma, environmental factors are considered relatively less important, though they can be used effectively.

## 1. Introduction

Injury is not just an “accident” [1,2]. This condition can be referred to as a disease. It has the necessary characteristics, such as type (e.g., blunt or penetrating), degree of severity, incidence, and prevalence [1]. Mortality associated with injury differs by nation or socioeconomic factors [3]. In addition, there are many prognostic systems, such as other diseases, an abbreviated injury scale (AIS), injury severity score (ISS), and trauma and injury severity score (TRISS) [4,5,6,7]. These prognostic systems mainly focus on patient status.

However, the biggest difference between other diseases and injuries is that the timing and mechanisms of the injury onset are clear [8]. Therefore, clinicians can easily predict the severity of injury using the information about its onset time and mechanisms (i.e., the magnitude and direction of the energy vector applied to the patient) [7]. The patient’s status has the greatest influence on the prediction of severity; however, the environment in which the injury occurs also have an influence [9]. These environmental factors include not only the circumstances in which patients were injured, but also the situation at the hospital (e.g., time of arrival).

Unlike other diseases, knowing the specific environmental factors related to the development of injury provides a huge advantage in prevention. It is a well-known fact that legal or cognitive improvement to reduce the occurrence of severe trauma involves much lower socioeconomic costs than the resources consumed by each individual with severe trauma [10,11,12,13]. In the past few decades, many countries focused on the prevention of severe trauma, and many legal and cognitive improvements were made (i.e., the prohibition of drunk driving and speeding, the mandatory wearing of safety equipment, the awareness of the duty to protect pedestrians, and the lowering of the speed limit in rain or snow conditions). Protective equipment for protecting against severe traumatic injuries also advanced significantly [14,15].

Although the legal, cognitive, and technological developments significantly lowered the mortality rate due to severe trauma, the proportion of environmental factors is minuscule in the system used for predicting the prognosis of traumatic diseases. Therefore, the purpose of this study was to analyze the influence of environmental factors on the survival of patients with severe trauma.

## 2. Materials and Methods

### 2.1. Study Population

This study was a retrospective analysis of the medical records of patients who were treated by the trauma team in the regional trauma center (RTC) of Gangwon Province, Republic of Korea, from 1 January 2016 to 31 December 2020. A total of 7108 patients activated the trauma team. Of these patients, younger individuals (<19 years old, *n* = 485), those with missing medical records or who were pronounced dead on arrival (*n* = 1183), mild trauma patients with an ISS < 16 (*n* = 3710), and TRISS-based non-preventable deaths (NP, TRISS < 25%) (*n* = 24) were excluded. Due to their significantly lower mortality rate than severe trauma patients (0.2% vs. 6.7%), mild trauma patients were not included in this analysis. In total, 1706 people were included in the study (Figure 1).

### 2.2. Trauma Team Activation

The trauma team at our institution consists of specialists in emergency medicine, surgery, thoracic surgery, neurosurgery, orthopedic surgery, and anesthesia. Regardless of duty time, the on-call team is made up of the same number of medical staff. Trauma teams, which are trained to maintain the same goals and course of treatment as a team, step in to provide resuscitation and treatment for severely injured trauma patients, starting with the initial assessment. The emergency medicine physician decides to activate the trauma team at the triage stage. The criteria for the activation of the trauma team are described in Table 1.

### 2.3. Data Collection

Patient demographic characteristics, including age, sex, time from injury to arrival, direct transportation (i.e., direct) or transferred via another hospital (i.e., indirect), previous medical history, ISS, AIS, intensive care unit (ICU) day, total hospital day, and vital signs at the time of initial arrival at the emergency department (systolic/diastolic blood pressure(sBP/dBP), pulse rate (PR), respiratory rate (RR), body temperature (BT), and Glasgow Coma Scale (GCS)), were obtained from medical records. The patient monitor system (CARESCAPE Monitor B650, GE Healthcare, Helsinki, Finland) automatically recorded sBP/dBP, PR, RR, and BT while the patient was lying on the bed. Using the GCS and ISS, we calculated the revised trauma score (RTS), as well as the TRISS, and divided the patients into definitively preventable (DP, TRISS > 50%) and potentially preventable (PP, TRISS 25–50%) groups. Arrival times were divided into three groups: day (transported to the RTC from 06:00 to 14:00), evening (transported to the RTC from 14:00 to 22:00), and night (transported on weekdays from 22:00 to 06:00, weekends). By extracting data from the website of the Korea Meteorological Administration, environmental factors, such as temperature, rainfall, snowfall, and humidity at the time of the accident, were collected for each patient. Laboratory results for hemoglobin (Hb) and delta neutrophil index (DNI) were also obtained.

### 2.4. Statistical Analysis

Continuous variables were expressed as mean ± standard deviation, and categorical variables were expressed as frequencies and percentages. Continuous data were tested for normal distribution using the Kolmogorov–Smirnov test, and compared using the Student’s *t*-test or the Mann–Whitney U test, as appropriate. A chi-squared test was used for categorical variables. A Cox proportional hazards regression analysis was performed to determine the relationship between prognostic factors and death within 30 days of the accident in all patients, direct patients, and indirect patients. Univariate and multivariate analyses were performed, and the variables were selected automatically using backward stepwise selection. Subgroup analyses for comparison of arrival times were performed after binary transformation of continuous variables. The non-linear relationships between the hazard ratio (HR; death within 30-days) and temperature, precipitation, snowfall, and humidity at the time of injury were evaluated using restricted cubic spline curves (RCS). In the RCS analyses, one or two cut-offs were determined via the changes in convexity of the prediction method. Statistical significance was set at *p* < 0.05. Statistical analysis was performed using R statistical software (version 4.1.0; R Foundation for Statistical Computing, Vienna, Austria).

## 3. Results

### 3.1. Patient Characteristics

This study included 1706 patients admitted between January 2016 and December 2020. A total of 115 patients died within 30 days, while 127 patients died in total (Figure 1). Among the 12 patients who died more than 30 days after the accident, the minimum and maximum times from accident to death were 31 and 381 days, respectively.

Patients were divided into two groups: >30-day survival (survival) and death within 30 days (death). Patients in the death group were significantly older than those in the survival group. There were no significant differences between the groups based on arrival time, direct/indirect, ICU day, DNI, systolic BP, diastolic BP, and RR. However, the death group had a significantly shorter time from injury to arrival, lower Hb level, higher PR, lower BT, lower GCS, and lower RTS than the survival group. More patients died within 30 days in the direct group and PP in the TRISS group. When comparing the degree of damage by body part, there were significantly more serious injuries (AIS ≥ 3) at head and neck, as well as facial and extremity injuries, in the death group. With regard to the environmental factors at the time of injury, there were no significant differences in temperature, snowfall, weather, and humidity. However, there was greater precipitation in the death group (Table 2).

### 3.2. Cox Proportional Hazards Regression Analysis for 30-Days Survival

In all patient groups, older age, high ISS, hypotension (systolic BP < 90 mmHg), and tachycardia (PR ≥ 100 beats/min) at arrival had harmful effects on 30-day survival. For environmental factors, high rainfall (hazard ratio [HR] 1.03, 95% confidence interval [CI] 1.00–1.05, *p* = 0.002) and night time (HR 1.72, 95% CI 1.02–2.92) had a negative effect on 30-day survival. Other environmental factors, such as humidity and temperature, had no significant effect on survival, and snowfall was excluded from the stepwise selection process. In contrast, the indirect group had a protective effect (HR 0.42, 95% CI 0.27–0.63) (Table 3A). Additional individual analyses were performed for the direct and indirect groups. In the direct group, night time was the only environmental factor that had a significant harmful effect on survival (HR 2.22, 95% CI 1.16–4.26) (Table 3B). However, in the indirect group, there was no significant effect from environmental factors (Table 3C). Age and ISS were variables that had significant effects on the three analyses.

### 3.3. Subgroup Analysis Based on Arrival Time

To evaluate the effect of the arrival time, subgroup analyses were performed after each variable was converted into a binary factor variable. In the evening versus day analysis, when it did not rain or snow (e.g., fine group), the evening time HR was 1.62 (95% CI, 0.95–2.74; *p* = 0.075), while when it rained or snowed (e.g., rain/snow group), the HR was 0.52 (95% CI, 0.22–1.24; *p* = 0.139). The interaction *p*-value of the two groups was 0.076, which was not statistically significant (Figure 2a). In contrast, in the analysis comparing night versus day, HR at night was 2.44 (95% CI, 1.30–4.58; *p* = 0.005) in the fine group (e.g., there was no rain or snow at the accident), while in the rain/snow group, the HR was 0.44 (95% CI, 0.15–1.31, *p* = 0.141). It was statistically significant that the interaction *p*-value between the two groups was 0.017 (Figure 2b). According to these findings, the mortality rate was higher at night and in the evening in good weather compared to rainy or snowy weather. The other subgroups did not show any significant results in either subgroup analysis.

### 3.4. Restricted Cubic Spline Curves for Continuous Variables

Weather-related environmental factors (i.e., temperature, humidity, rainfall, and snowfall) were continuous variables, and the previous analyses were performed through assuming linearity. Consequently, an additional RCS analysis was performed to study the linear or non-linear relationships between these variables and mortality. Temperature had an inverted U-shaped relationship with mortality. Two cut-offs were determined as 2.5 °C and 24.0 °C (Figure 3a). Humidity had a U-shaped relationship, and two cut-offs of 42 and 79% were determined (Figure 3b). Furthermore, rainfall (Figure 3c) and snowfall (Figure 3d) had a relatively linear relationship, and the respective cut-offs were 7.4 mm and 1.9 cm.

## 4. Discussion

Trauma is one of the leading causes of death worldwide [1,2,3,4,5,6]. However, unlike other chronic or acute diseases, trauma is a straightforward phenomenon; therefore, its perception is relatively vague. The fact that the onset mechanism and time of injuries are clearly established is the most significant characteristic of trauma compared to other diseases. Knowing when a specific disease occurs makes it possible to search for conditional factors in which the disease may occur or become more severe. Therefore, to estimate the prognosis in patients with severe trauma, the environmental factors of trauma should be considered. However, the proportion of environmental factors in current trauma prognosis prediction systems is minimal. Differentiating between penetrating or blunt injuries is the only environmental factor included in the widely used prognostic prediction systems [4,5,6,16,17,18].

However, there are some considerations in adding environmental factors to prognostic systems. Firstly, there are too many environmental variables at the accident site, and these environmental factors vary depending on the mechanism (e.g., in the case of a traffic accident, the type and speed of the vehicle and the location of the patient in the vehicle are environmental factors, whereas in the case of a fall accident, the height, type of floor, and safety equipment are environmental factors.) [9]. Therefore, in this study, the common environmental factors were selected for each mechanism of injury. Secondly, investigation into environmental factors is usually neglected because of the urgency and rapid progression of severe trauma patients. Therefore, in this study, environmental factors were investigated using an automated crawling technique applied on the Korea Meteorological Administration’s website through the location and time of each patient’s injury. Thirdly, the complexity of multiple traumas with a wide spectrum of severity, even if the same mechanism is involved, makes it difficult to consider environmental factors. Consequently, in this study, the effect of severity change was corrected through calculating the TRISS score, excluding NP, and adding ISS to the statistical model.

It is well known that preventing the occurrence of severe trauma is much more advantageous, given that the socioeconomic costs of managing severe trauma patients are enormous [10,11,12,13,14,15,19,20]. Accordingly, many studies on the influence of environmental factors on the occurrence of severe trauma were conducted from social and institutional perspectives [21,22,23]. However, only a few studies analyzed survival rates from a medical perspective. Environmental factor variables, such as rainfall and arrival time, which were not significant in the univariate analysis, showed significant effect in multivariate Cox regression analysis, and demonstrated a linear relationship with HR in additional RCS analysis; that is, more patients died due to higher rainfall or snowfall. The HR of the indirect group was significantly lower than that of the direct group, which cannot exclude the possibility of selection bias (i.e., early mortality cases may have been excluded from the indirect group). Therefore, an additional Cox regression analysis was performed, considering the transfer status. In the direct group, HR was increased in both the night and evening, though it was not significant in the evening. The significant increase in HR at night seems to be due to the fatigue of the medical staff and insufficient resource availability compared to during the day. Therefore, these results raise the need to consider fatigue and redistribution of resources in trauma-related medical institutions. In the indirect group, no significant effects of environmental factors were observed. This finding seems to be due to the resuscitation effects at a previous medical institution and the exclusion of early mortality cases in the indirect group. Additional subgroup analyses for arrival times were performed. When there was no rain or snow at the time of the accident (e.g., fine), HR was increased significantly at night (*p* = 0.005), while in the evening, HR was increased, albeit not significantly (*p* = 0.075). The difference between the fine and rain/snow groups at night was significant (*p* = 0.017). This result seems to be due to the tendency of individuals to be more vigilant in rainy or snowy weather, when there is no natural light.

In the RCS analysis to confirm the linearity or non-linearity of continuous variables, U- and inverted U-shaped relationships were confirmed in temperature and humidity, which were not significant in multivariate Cox analysis. The Cox regression model assumes the linearity of the covariates in the log hazard function. This model assumes that linearity is the most used statistical technique to show the relationship between the dependent and independent variables; it also assumes that linearity in a continuous variable that exhibits nonlinearity is likely to underfit the actual influence of the variable [24,25,26].

The limitations of this study are as follows. To select common environmental factors among the various mechanisms of trauma, only a few environmental factors could be selected. Due to the limited availability of regional weather data, information on sensible temperature and wind speed was also unavailable. Additional research should provide more details on the connection between hypothermia and the working circumstances of pre-hospital first responders. The penetrating injuries were excluded before the analyses because the proportion of penetrating injuries in Korea is extremely low, currently being approximately 3.5–9.2% [3,27]. Since this was a single-institution study, the possibility of selection bias cannot be ruled out. Consequently, additional research should be conducted with individual analysis for each mechanism of trauma at a multi-center study, and with a larger cohort, including local medical emergency centers or agencies.

## 5. Conclusions

The mortality of severe trauma patients can be predicted not only through assessment the patient’s medical problems, but also through various environmental factors. As a result of the findings of this study, which showed that a variety of environmental factors such as temperature, weather, or duty time had an effect on the 30-day survival of severely injured patients, it is suggested that preventative measures, such as additional staffing, be taken to offset environmental factors that are anticipated to increase trauma patients’ mortality. New survival prediction models that integrate environmental and clinical data from prospective multicenter studies may help to achieve these goals.

## Figures and Tables

**Figure 1 healthcare-11-01333-f001:**
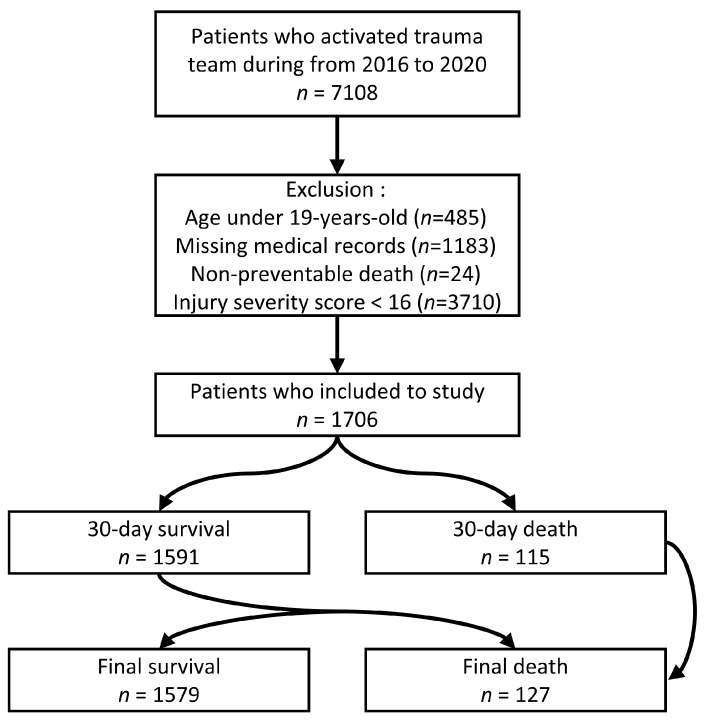
Flow-chart of this study.

**Figure 2 healthcare-11-01333-f002:**
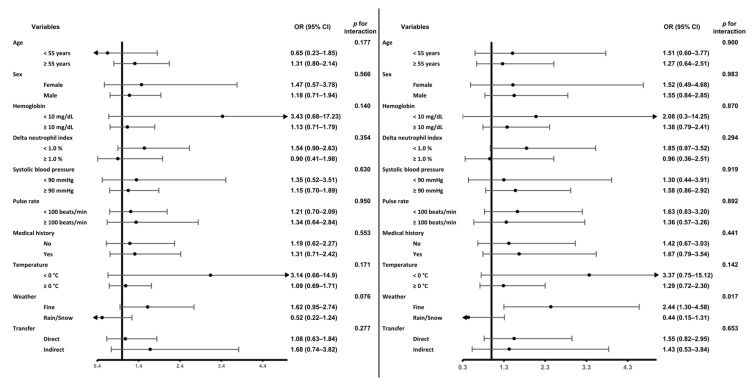
Subgroup analysis based on arrival time. (**left**) Comparison between evening and day times. (**right**) Comparison between night and day times.

**Figure 3 healthcare-11-01333-f003:**
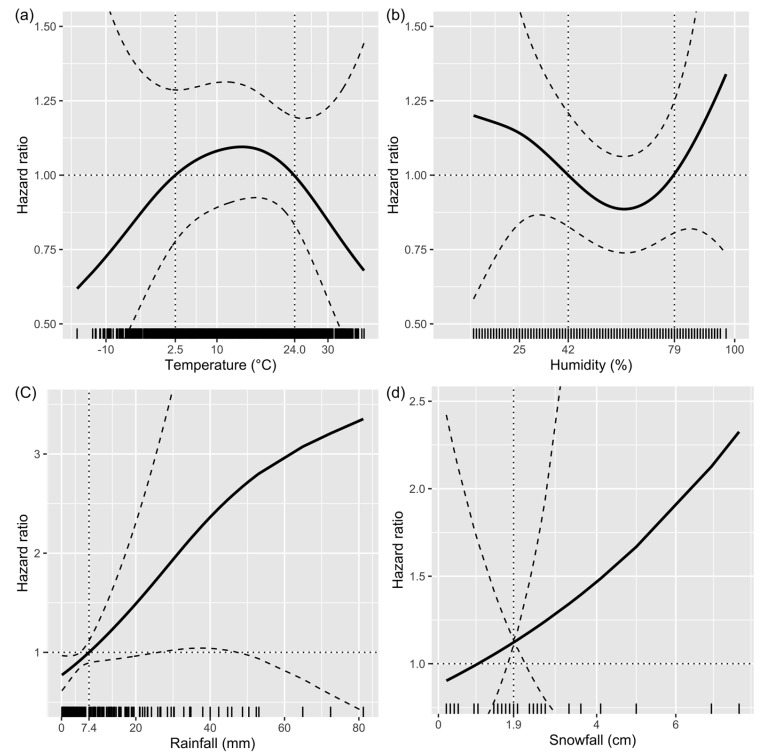
Restricted cubic spline curves of continuous variables. (**a**) Temperature in accident location. (**b**) Humidity in accident location. (**c**) Rainfall at time of accident. (**d**) Snowfall at time of accident.

**Table 1 healthcare-11-01333-t001:** Criteria for activation of trauma team. If any of following conditions meet criteria, a call to trauma team is made.

1. Physiology
A. Airway obstruction/respiratory distress
B. Intubation state
C. Respiratory rate < 10/min or >29/min
D. Systolic blood pressure <90 mmHg for older than 10 years old <70 + 2(Age) mmHg for 1–10 years old <60 mmHg for under than 1 year old
E. Glasgow Coma Scale ≤ 13, pain response, or unconsciousness
F. A patient transferred with a blood transfusion to maintain vital signs
G. Deterioration of condition in stable patient
2. Anatomical
A. Penetrating injury
i. Penetrating injury of head and neck, chest, or abdomen
ii. Limbs: Penetrating injury of elbow or above knee
B. Chest
i. Flail chest
C. Neurological
i. Opened or inverted skull fracture
ii. Quadriplegia or suspected spinal cord injury
D. Orthopedic
i. Pelvic bone fracture
ii. Fractures to two or more proximal long bones
iii. Crushed/flawed/cleaved damage to limbs or loss of pulse
iv. Amputation of upper proximal wrist or ankle
3. Mechanisms of trauma
A. A patient in a traffic accident in which the passenger died
B. A patient who fell out of a vehicle during a traffic accident
C. Traffic accidents over 60 km/h
D. Accidents between vehicles and pedestrians exceeding 30 km/h
E. It takes more than 20 min to rescue the patient (the vehicle is pressed ≥ 30 cm)
F. Motorcycles, bicycles, and other vehicles: Collision or overturning at 30 km/h or more
G. Falls over 6 m for adults and 3 m for children
H. Explosion
4. Other cases which the doctor resident in the resuscitation area determines to be necessary

**Table 2 healthcare-11-01333-t002:** Demographic characteristics of enrolled patients.

Characteristics	All Patients	30-Day Survival	30-Day Death	*p* Value
	(*n* = 1706)	(*n* = 1591)	(*n* = 115)	
Age	57.8 ± 16.0	57.2 ± 15.8	65.4 ± 17.2	<0.001
Sex				1.000
Female	415 (24.3%)	387 (24.3%)	28 (24.3%)	
Male	1291 (75.7%)	1204 (75.7%)	87 (75.7%)	
ICU day	9.3 ± 16.5	9.5 ± 16.9	6.8 ± 6.6	0.708
Hemoglobin (mg/dL)	13.0 ± 2.1	13.1 ± 2.0	12.3 ± 2.3	<0.001
Delta neutrophil index (%)	1.3 ± 2.6	1.3 ± 2.7	1.1 ± 1.8	0.099
Past medical history	733 (43.0%)	672 (42.2%)	61 (53.0%)	0.031
Systolic blood pressure (mmHg)	130.1 ± 32.1	129.8 ± 31.1	134.3 ± 43.4	0.098
Diastolic blood pressure (mmHg)	75.9 ± 18.9	76.1 ± 18.6	73.5 ± 23.5	0.444
Pulse rate (beats/min)	87.4 ± 20.0	86.9 ± 19.2	94.9 ± 28.1	0.003
Respiratory rate (beats/min)	19.7 ± 2.4	19.7 ± 2.3	20.3 ± 3.6	0.078
Body temperature (°C)	36.4 ± 0.8	36.4 ± 0.8	36.0 ± 0.9	<0.001
Injury severity score	22.2 ± 6.1	21.8 ± 5.9	27.0 ± 7.1	<0.001
Glasgow Coma Scale	13.3 ± 3.0	13.6 ± 2.7	9.6 ± 4.1	<0.001
Revised trauma score	7.4 ± 0.9	7.4 ± 0.8	6.3 ± 1.2	<0.001
Trauma and injury severity score				<0.001
Definitely preventable (>50%)	1637 (96.0%)	1549 (97.4%)	88 (76.5%)	
Potentially preventable (20~50%)	69 (4.0%)	42 (2.6%)	27 (23.5%)	
Severe injury by body part				
Head and neck	832 (48.8%)	751 (47.2%)	81 (70.4%)	<0.001
Face	24 (1.4%)	19 (1.2%)	5 (4.3%)	0.018
Chest	784 (46.0%)	736 (46.3%)	48 (41.7%)	0.399
Abdomen	346 (20.3%)	328 (20.6%)	18 (15.7%)	0.247
Extremities and pelvis	428 (25.1%)	409 (25.7%)	19 (16.5%)	0.037
Others	29 (1.7%)	29 (1.8%)	0 (0.0%)	0.277
Holiday	580 (34.0%)	540 (33.9%)	40 (34.8%)	0.935
Temperature (°C)	15.1 ± 10.4	15.1 ± 10.5	14.9 ± 9.5	0.807
Rainfall (mm)	7.7 ± 12.1	7.3 ± 11.8	12.1 ± 14.2	0.031
Snowfall (cm)	1.9 ± 1.7	1.9 ± 1.6	2.3 ± 2.4	1.000
Weather				0.809
Fine	1332 (78.1%)	1245 (78.3%)	87 (75.7%)	
Rain	334 (19.6%)	309 (19.4%)	25 (21.7%)	
Snow	40 (2.3%)	37 (2.3%)	3 (2.6%)	
Humidity (%)	55.9 ± 21.3	55.9 ± 21.2	55.7 ± 22.6	0.861
Arrival time				0.717
Day	544 (31.9%)	511 (32.1%)	33 (28.7%)	
Evening	810 (47.5%)	754 (47.4%)	56 (48.7%)	
Night	352 (20.6%)	326 (20.5%)	26 (22.6%)	
Time from injury to arrival (min)	297.2 ± 1197.1	309.1 ± 1237.3	132.6 ± 223.9	<0.001
Transfer				0.002
Direct	868 (50.9%)	793 (49.8%)	75 (65.2%)	
Indirect	838 (49.1%)	798 (50.2%)	40 (34.8%)	

**Table 3 healthcare-11-01333-t003:** Univariate and multivariate Cox proportional hazard regression analysis of risk factors for 30-day survival. (**A**) For all enrolled patients. (**B**) For patients who transferred directly from accidental place (e.g., direct). (**C**) For patients who transferred via another hospital (e.g., indirect).

(A)				
Characteristics	Univariate		Multivariate	
	HR (95% CI)	*p*	HR (95% CI)	*p*
Age	1.04 (1.02–1.05)	<0.001	1.04 (1.03–1.06)	<0.001
Gender				
Female	Reference		Reference	
Male	1.02 (0.67–1.57)	0.916	1.51 (0.95–2.41)	0.081
Rainfall	1.02 (1.00–1.04)	0.029	1.03 (1.00–1.05)	0.020
Snowfall	1.08 (0.73–1.59)	0.709	0.98 (0.63–1.53)	0.938
Humidity	1.00 (0.99–1.01)	0.85	1.00 (0.99–1.01)	0.791
Temperature	1.00 (0.98–1.02)	0.943	0.99 (0.98–1.01)	0.592
Arrival time				
Day	Reference		Reference	
Evening	1.16 (0.76–1.79)	0.494	1.38 (0.86–2.22)	0.183
Night	1.24 (0.74–2.07)	0.418	1.72 (1.02–2.92)	0.044
Injury severity score	1.10 (1.07–1.12)	< 0.001	1.10 (1.07–1.12)	<0.001
Hemoglobin	0.87 (0.80–0.94)	< 0.001	0.91 (0.82–1.01)	0.083
DNI	0.93 (0.84–1.04)	0.219	0.93 (0.83–1.05)	0.250
Systolic BP				
≥90 mmHg	Reference		Reference	
<90 mmHg	2.88 (1.89–4.38)	<0.001	1.62 (1.01–2.62)	0.047
Pule rate				
<100 beats/min	Reference		Reference	
≥100 beats/min	1.97 (1.35–2.88)	<0.001	2.35 (1.58–3.49)	<0.001
Transfer				
Indirect	Reference		Reference	
Direct	0.51 (0.35–0.75)	<0.001	0.42 (0.27–0.63)	<0.001
(**B**)				
**Characteristics**	**Univariate**		**Multivariate**	
	**HR (95% CI)**	** *p* **	**HR (95% CI)**	** *p* **
Age	1.02 (1.01–1.04)	0.002	1.03 (1.01–1.05)	<0.001
Gender				
Female	Reference		Reference	
Male	1.26 (0.70–2.25)	0.436	2.05 (1.09–3.85)	0.026
Rainfall	1.02 (0.99–1.04)	0.187	1.02 (0.99–1.05)	0.142
Humidity	1.00 (0.99–1.02)	0.448	1.00 (0.99–1.01)	0.920
Temperature	0.99 (0.97–1.01)	0.262	0.98 (0.96–1.01)	0.193
Arrival time				
Day	Reference		Reference	
Evening	0.98 (0.58–1.65)	0.932	1.34 (0.75–2.39)	0.326
Night	1.53 (0.84–2.81)	0.168	2.22 (1.16–4.26)	0.016
Injury severity score	1.11 (1.08–1.14)	<0.001	1.11 (1.07–1.14)	<0.001
Hemoglobin	0.81 (0.72–0.90)	<0.001	0.85 (0.74–0.97)	0.020
DNI	0.82 (0.66–1.01)	0.065	0.76 (0.60–0.95)	0.014
Systolic BP				
≥90 mmHg	Reference		Reference	
<90 mmHg	2.86 (1.69–4.87)	<0.001	1.30 (0.70–2.43)	0.410
Pule rate				
<100 beats/min	Reference		Reference	
≥100 beats/min	2.50 (1.57–3.99)	<0.001	2.94 (1.79–4.83)	<0.001
(**C**)				
**Characteristics**	**Univariate**		**Multivariate**	
	**HR (95% CI)**	** *p* **	**HR (95% CI)**	** *p* **
Age	1.07 (1.04–1.09)	<0.001	1.07 (1.04–1.11)	<0.001
Gender				
Female	Reference		Reference	
Male	0.69 (0.36–1.31)	0.255	1.05 (0.51–2.15)	0.897
Rainfall	1.03 (0.99–1.06)	0.112	1.03 (1.00–1.07)	0.091
Humidity	0.99 (0.98–1.00)	0.191	0.99 (0.98–1.01)	0.463
Temperature	1.02 (0.99–1.05)	0.277	1.02 (0.99–1.05)	0.285
Arrival time				
Day	Reference		Reference	
Evening	1.83 (0.82–4.08)	0.137	1.63 (0.66–4.02)	0.289
Night	1.19 (0.45–3.17)	0.727	1.22 (0.44–3.38)	0.708
Injury severity score	1.08 (1.03–1.12)	< 0.001	1.08 (1.04–1.13)	<0.001
Hemoglobin	0.83 (0.72–0.95)	0.007	1.02 (0.86–1.20)	0.846
Delta neutrophil index	1.01 (0.96–1.08)	0.623	1.02 (0.96–1.08)	0.481
Systolic blood pressure				
≥90 mmHg	Reference		Reference	
<90 mmHg	3.07 (1.53–6.14)	0.002	2.19 (1.00–4.79)	0.049
Pule rate				
<100 beats/min	Reference		Reference	
≥100 beats/min	1.56 (0.82–2.99)	0.178	1.76 (0.91–3.41)	0.095

HR, hazard ratio; CI, confidence interval; DNI, Delta neutrophil index; BP, Blood pressure.

## Data Availability

The data presented in this study are available on request from the corresponding author. The data are not publicly available to protect the privacy of enrolled patients.

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
