# Peer review of "Multifaceted Analysis of the Environmental Factors in Severely Injured Trauma: A 30-Day Survival Analysis"

_healthcare, 2023, doi:10.3390/healthcare11091333_

Round 1

Reviewer 1 Report

Review

The authors noticed that the proportion of environmental factors used in the system for predicting the survival prognosis after trauma is minimal. Therefore, the main goal of the submitted article was to analyze the influence of environmental factors on the survival among patients with severe trauma. Another aim was the development of the integrated survival prediction model that combines environmental and patient related factors. Third aim was to propose some new strategies to improve survival of trauma patients on basis of performed analysis.

There were  1706 trauma patients included in the study. The main results were that night compared with day and high rainfall were significant environmental predictors of mortality due to severe trauma. There were linear relationship between mortality and precipitation and inverted U-shaped relationship between mortality and air temperature. The main conclusion was that environmental factors can be effectively used in predicting the survival of patients with severe trauma. The analysis of the impact of environmental factors on survival after severe trauma is interesting and might have very practical influence on clinical practice and survival predicting systems that should be developed in the future. 

In my opinion, the article is publishable, but needs major improvements:

Introduction:

Line 50 – please change “users” into some other word because it is not clear if it concerns patients or medical staff taking care of patients.

Materials and Methods

Line 70-71 – please precisely specify the differences (if they exist) in the number of medical staff (doctors, nurses, paramedics) involved in the trauma team during different hours of a day (6-14; 14-22; 22-6) and weekends.

Table 1.

Please provide the information how many of the presented criteria have to be present to activate the trauma team, one? more than one? One/two from each group?

Line 80-81 – please provide the information what does it mean that the vitals were collected from medical records. Does it mean that those vitals were present at the arrival to the emergency department or those vitals were first that were measured by the pre-hospital team?

Line 87-88 – please explain why the information about the wind was not collected. In my opinion air temperature and wind speed should be analyzed together in the form of wind chill temperature. That would give more complete information about environmental risk of hypothermia and working environment of pre-hospital team. Adding wind speed and to calculate wind chill temperature would be a  more complete analysis of environmental factors potentially affecting the survival after trauma. If it is possible please perform such analysis, if not write about it in the Limitations section.

Results

Line 121 – what is PP?

Lines 151-153 – Please rephrase this sentence because it is difficult to understand.

Figure 1 – please improve the arrows. In my opinion there should be only one arrow coming from (30day death N 115) to (Final death N 127)  - please cancel arrow to (Final Survival N 1579). There should two arrows from (30 day survival N 1591) – first into (Final survival N 1579) and second into (Final death N 127)

Discussion

Line 196 – I would argue if the environmental factors are “essential”. I would rather suggest to write that they should be considered.

Conclusions

Data presented in the study  do not support all the conclusions. There is no evident proof presented in the study that would support the conclusion that environmental factors can be used effectively in the predicting the survival of patients with severe trauma. To conclude above the prospective study should be performed. In my opinion the authors should improve the conclusions which in the current version of the article are very general and not fully supported by the results.

The authors should comment in the conclusion section on all of the aims presented in the introduction section. The main goal of the submitted article was to analyze the influence of environmental factors on the survival among patients with severe trauma. This aim was fulfilled  - it should be commented in conclusions. Another aim was the development of the integrated survival prediction model that combines environmental and patient related factors – it was not done in the study – I suggest to resign from this aim as it was not analyzed. Third aim was to propose some new strategies to improve survival of trauma patients on basis of performed analysis – the authors should propose those new strategies that could be implemented to improve survival of the patients. Such list would be beneficial for emergency medicine doctors and paramedics who work pre-hospital and at the emergency departments.

Kind regards

Author Response

Thank you for your good comments. And we appreciated that we take an opportunity to revise this manuscript according to your comments. Your efforts will make my research more fruitful for the future. And I enjoyed this process. Thanks reviewers and Editer in Chief for giving me this opportunity.

Reviewer 1

Introduction:

Line 50 – please change “users” into some other word because it is not clear if it concerns patients or medical staff taking care of patients.

Answer: Thank you for your kind comment and recommendation. As indicated by the yellow marker in the text, we have changed the wording to "Protective equipment for protecting against severe traumatic injuries has also advanced significantly." (Line 50-51)

Materials and Methods

Line 70-71 – please precisely specify the differences (if they exist) in the number of medical staff (doctors, nurses, paramedics) involved in the trauma team during different hours of a day (6-14; 14-22; 22-6) and weekends.

Answer: Thank you for your comment. The number of medical staff has remained constant regardless of duty time because the trauma team at our institution has been trained to share common goals and processes of care. This is discussed in more detail on lines 68–72.

Table 1. Please provide the information how many of the presented criteria have to be present to activate the trauma team, one? more than one? One/two from each group?

Answer: Thank you for your comment. If any of the criteria in Table 1 are met, the trauma team is activated. This is noted in the table title. (Line 75-76)

Line 80-81 – please provide the information what does it mean that the vitals were collected from medical records. Does it mean that those vitals were present at the arrival to the emergency department or those vitals were first that were measured by the pre-hospital team?

Answer: Thank you for your kind comment. We apologize for the lack of explanation of this information. These vital sign information were verified as soon as the patient arrived at the emergency room. This is discussed in more detail on lines 79–86.

Line 87-88 – please explain why the information about the wind was not collected. In my opinion air temperature and wind speed should be analyzed together in the form of wind chill temperature. That would give more complete information about environmental risk of hypothermia and working environment of pre-hospital team. Adding wind speed and to calculate wind chill temperature would be a more complete analysis of environmental factors potentially affecting the survival after trauma. If it is possible please perform such analysis, if not write about it in the Limitations section.

Answer: Thank you for your kind comment. We totally agree with your viewpoint. Your suggestions regarding the sensible temperature and wind speed were taken into consideration, but due to the lack of weather data, we were forced to leave them out of the study. We are planning for a larger prospective study in this area with more environmental data. This limitation was added to the discussion. (Line 257-260)

Results

Line 121 – what is PP?

Answer: Thank you for your comment. The PP group, which is mentioned on line 88 earlier, stands for potentially preventable death group.

Lines 151-153 – Please rephrase this sentence because it is difficult to understand.

Answer: Thank you for your recommendation. We apologize for the clarity of our explanation. For the purpose of clarity, these sentences have been revised. (Line 159-162)

Figure 1 – please improve the arrows. In my opinion there should be only one arrow coming from (30day death N 115) to (Final death N 127)  - please cancel arrow to (Final Survival N 1579). There should two arrows from (30 day survival N 1591) – first into (Final survival N 1579) and second into (Final death N 127)

Answer: Thank you for mentioning such an important point. As mentioned, figure 1 was revised.

Line 196 – I would argue if the environmental factors are “essential”. I would rather suggest to write that they should be considered.

Answer: Thank you for making this important point and your kind recommendation. We agreed your recommendation, we revised this sentence as you mentioned. (Line 203-205)

Conclusions

Data presented in the study  do not support all the conclusions. There is no evident proof presented in the study that would support the conclusion that environmental factors can be used effectively in the predicting the survival of patients with severe trauma. To conclude above the prospective study should be performed. In my opinion the authors should improve the conclusions which in the current version of the article are very general and not fully supported by the results.

The authors should comment in the conclusion section on all of the aims presented in the introduction section. The main goal of the submitted article was to analyze the influence of environmental factors on the survival among patients with severe trauma. This aim was fulfilled  - it should be commented in conclusions. Another aim was the development of the integrated survival prediction model that combines environmental and patient related factors – it was not done in the study – I suggest to resign from this aim as it was not analyzed. Third aim was to propose some new strategies to improve survival of trauma patients on basis of performed analysis – the authors should propose those new strategies that could be implemented to improve survival of the patients. Such list would be beneficial for emergency medicine doctors and paramedics who work pre-hospital and at the emergency departments.

Answer: We appreciated your important viewpoints and suggestions and agreed with them. We humbly acknowledge that we did not meet the purpose we outlined in the introduction. As a result, we have removed purposes 2 and 3 from the introduction and changed the conclusion to state that our study shows that environmental factors are related to mortality in patients with severe trauma. Instead, as was already mentioned, we are currently planning a prospective, multicenter study in the region, and we proposed the potential of a novel model integrate clinical and environmental factors that could decrease mortality in severe trauma patients through further research in conclusion. We have also added example of interventions to address factors that adversely affect patient mortality. (Line 270-276)

Reviewer 2 Report

good afternoon!

In the statistical part, it is not possible to carry out the Shapiro Will test when there are more than 50 people, the correct test would be that of Kolmogorov. (LINE 95).

For the rest I think it's fine! I think it would be correct to carry out the normality test again for saebr if one test or another is used.

clean English that is understood very well

Author Response

Thank you for your good comments. And we appreciated that we take an opportunity to revise this manuscript according to your comments. Your efforts will make my research more fruitful for the future. And I enjoyed this process. Thanks reviewers and Editer in Chief for giving me this opportunity.

Reviewer 2

In the statistical part, it is not possible to carry out the Shapiro Will test when there are more than 50 people, the correct test would be that of Kolmogorov. (LINE 95).

For the rest I think it's fine! I think it would be correct to carry out the normality test again for saebr if one test or another is used.

Answer: We appreciate your kind and important comment. We have taken your recommendation very seriously and have reworked the statistics as you suggested. Fortunately, the results were not changed. We will keep this in mind for future studies.

interventions to address factors that adversely affect patient mortality. (Line 270-276)

Reviewer 3 Report

The manuscript by Dr. Chung and colleagues is a retrospective analysis of the patient charts for the environmental factors in the consideration of trauma. The manuscript is based on an interesting concept of combined efforts of the healthcare team with Korea Meteorological Administration. I commend the authors for the study and including the detailed thoughts about the environmental factor role and its impact in trauma. There are few considerations that need to be addressed:

1. Title should include severe trauma since mild trauma is excluded.

2.Table 1 mention demographics of 5416 patients however, figure 1 mentions that 1706 patients met the inclusion criteria. This is a major discrepancy. 

3. The criteria for trauma team activation is well presented, however, it was difficult to know if the management done is different if the trauma team activation is not done if the criteria for speed are not met? Kindly clarify with more details why patients with mild trauma were excluded from the study since environmental factors could play a role in that too.

4. Conclusion of the study is very abrupt. It would be better if authors can mention which environmental factors specifically should be considered in the outcome. Last sentence of the conclusion-" Consequently, preventive measures in an environmental situation where the mortality rate in trauma patients is expected to be high" - seems incomplete.

Author Response

Thank you for your good comments. And we appreciated that we take an opportunity to revise this manuscript according to your comments. Your efforts will make my research more fruitful for the future. And I enjoyed this process. Thanks reviewers and Editer in Chief for giving me this opportunity.

Reviewer 3

  1. Title should include severe trauma since mild trauma is excluded.

Answer: Thank you for your kind recommendation. We revised the title as “Multifaceted Analysis of the Environmental Factors in Severely Injured Trauma: A 30-day Survival Analysis”

2.Table 1 mention demographics of 5416 patients however, figure 1 mentions that 1706 patients met the inclusion criteria. This is a major discrepancy.

Answer: Thank you for your important comment. We apologize for errors in the numbers. This was a very serious mistake, and we have confirmed that the study was actually conducted with data from 1706 patients and have corrected it.

  1. The criteria for trauma team activation is well presented, however, it was difficult to know if the management done is different if the trauma team activation is not done if the criteria for speed are not met? Kindly clarify with more details why patients with mild trauma were excluded from the study since environmental factors could play a role in that too.

Answer: This study was conducted in a patient population where a trauma team was activated. The role of the trauma team is newly described on lines 68-72. Because trauma teams are trained with the common goal of treating patients with severe trauma, there are differences in care when a trauma team is activated and when it is not. Because of their significantly lower mortality rate than severe trauma patients (0.2% vs. 6.7%), mild trauma patients were not included in this analysis. (Line 65-67)

  1. Conclusion of the study is very abrupt. It would be better if authors can mention which environmental factors specifically should be considered in the outcome. Last sentence of the conclusion-" Consequently, preventive measures in an environmental situation where the mortality rate in trauma patients is expected to be high" - seems incomplete.

Answer: Thank you for your kind comment and recommendation. We agree with you that the conclusion of this study is very abrupt. We have removed purposes that were unrelated to the conclusion in the introduction and have changed the conclusion to state that our study shows that environmental factors are related to mortality in patients with severe trauma. We proposed the potential of a novel model integrate clinical and environmental factors that could decrease mortality in severe trauma patients through further research in conclusion. We have also added example of interventions to address factors that adversely affect patient mortality. (Line 270-276)

Round 2

Reviewer 3 Report

The manuscript reads well and I think it is acceptable for publication